# Marvel Presents a Global Utopia and Confronts Nationalism: Eternals as a New Mythology Forged from Western Roots

**Emma de Beus**

Independent Researcher, New York, NY 10032, USA; eedebeus@gmail.com

**Abstract:** Marvel's 2021 film *Eternals* presents a new mythology for a new century, for an audience grappling with the complexity of postcolonialism and concerned about resurging white nationalism. Its mythology, while rooted in Western narratives, presents a utopia in the form of a multicultural pantheon, presented by a carefully selected, diverse class. While Marvel undoubtedly has commercial concerns, its careful construction of this new mythology and the considered adaptation process show a moral vision for the future. Importantly, this vision presents a direct contrast to the resurgence of the appropriation of classical mythology as justification for white supremacy. Marvel's *Eternals* therefore can be seen as utopian: it offers the perfection of moral predictability, of good triumphing over evil. However, it simultaneously undercuts its story by couching it in the genre of a comic book superhero fantasy adventure–the reality *Eternals* offers, even fictionally, is beyond ordinary, mortal humans.

**Keywords:** mythology; utopian; superhero; nationalism; postcolonialism

## 1. Introduction

Marvel's 2021 film *Eternals*, directed by Chloé Zhao (Zhao 2021), cherry-picks pieces of Western canonical mythological, historical, and literary traditions, assembles a global cast whose nationalities or heritage can be read as comments on a postcolonial world, and then compiles these elements into a new, highly politicized form of myth. Roland Barthes and Joseph Campbell have theorized the notion of a new mythology as a way to imagine ancient models fitting into modern society. Marvel introduces this tension in *Eternals* in that the Western literary and historical narrative it uses is inherently exclusionary, but the chosen actors present a more global perspective, enabling the film to reclaim some of the individual power taken away by nationalism, imperialism, and colonialism.

*Eternals* presents a 21st century mythology, and through its adaptation of Western narratives and storytelling choices, urges the audience to see its worth through a utopian lens. Here, "utopia" will be considered both to mean a perfect, ideal place and also one that is inherently impossible, as no reality can be truly perfect. In the utopia offered by *Eternals*, all facets of history, colonialism, and racism are present and reality is reconfigured as a global nation, replacing the disparate national polarization of today.

However, inviting the audience to see the story, characters, and source material offered here as a potential utopia is paradoxically problematic. How can the root of all civilization be Ancient Greece? How can the values inherent to Western literature and mythology house the diversity needed to encompass the full breadth of human experience and beliefs? Thinking about the film as a utopia forces the audience to examine the issues of race, nationalism, and colonialism in historical, literary, and present-day contexts and ultimately to contrast today with the film's utopia.

For the purposes of this analysis, the audience is an uncomfortable blend of monolithic and diasporic. When box office takings are measured—the universal answer to "will there be a sequel?"—they are divided into two sections: domestic and international. While certain markets are more prominent within that second category, there is clearly a bias

towards the input of American audiences. However, the nature of this film, as will be discussed, is such that particular care is taken to diversify the film and by extension its target audience.

It is important to note that prior to the 2021 film *Eternals*, the comic book form has already filtered through various literary, historical, and mythological texts to create these characters and the basic premises of various story arcs for them. This initial adaptation of classical sources offers a Western-centric narrative. The secondary adaptation taking place in *Eternals* serves as a postcolonial response to both the comic book decisions and the narratives on which they draw. Consequently, Marvel films such as this one need to be considered through the intermediary of their comic books. The *Eternals* characters first appeared in July 1976 (DeFalco et al. 2019, p. 125) and it is therefore vital to understand that decades of comic book source material were available when this film was conceptualized and plotted. The creative minds behind this film had far more source material than they could ever fit into one film, so it is particularly important to track what storylines they chose to tell with these characters as they must be interpreted as deliberate artistic, nuanced choices rather than as ones dictated by the source material.

The film focuses on ten eternal beings who have been a part of Earth's history and cultural generation since the beginning of humanity. Their role is not to interfere with human events or experience such as war or technological advances, but rather to eradicate another alien species, the Deviants, from the planet. The issue of whether or not this vision of history is both Western and non-Western recurs in different ways throughout the film, challenging the Western canon's relationship with nationalism and race. Although the film starts in the present day, flashbacks spaced throughout the film show the important roles these characters have played throughout history. Through these flashbacks, the audience is sped through human history from 5000 BCE with the arrival of the Eternals–who are sent by the even more powerful Arishem, who is a Celestial–to 1521 CE, when apparently the Deviants are eradicated from the planet. This particular date is key for seeing the characters and the film directly respond to Western colonialism. The year 1521 CE sees the Eternals eradicate the Deviants in the same time and place as the Spanish conquistadors are committing genocide against the indigenous people of the Aztec capital city of Tenochtitlan. This moment in human history fractures the group: some want to prevent the genocide while some say they must stick to the parameters of their mission, which dictate that they have to stay out of human conflict unless Deviants are involved.

Although the group is not allowed to leave Earth until Arishem permits them to do so, they spend the next five hundred years disbanded, living, for the most part, individual lives at least in part because of this moral disagreement. Certainly, the fracturing of the group is also caused both by the seeming completion of their shared mission, but of all the moments in over 6500 years of human history that could have driven the group apart, the film chooses one of history's worst moments of colonialism, nationalism, and racism on which to center the plot. The film then more or less smash-cuts to today, when not only are the Deviants actually not eradicated, but the moral, social, and political issues the Eternals debated in 1521 CE are still much in evidence.

Ultimately, Marvel's 2021 film *Eternals* presents a new mythology, still constructed from a Western-centric canon, but enacted as a utopian, multicultural pantheon.

## 2. Literature Review

There is a vocal contingent of scholars arguing that it is time to "build critical momentum" in order to reach broader consensus regarding "the cultural significance of comics as an aesthetically complex, historically rich, and substantially American medium" (Stein et al. 2011, p. 502). Some specifically focus on the ways in which stigma surrounding this medium has "significantly impeded the evolution of the comic book as an art form" (Lopes 2006, p. 387). Both of these perspectives influence this discussion of *Eternals*. The ability this art form possesses, particularly in its adapted film form, to speak to today's issues of nationalism, racial polarization, and the postcolonial impact of both of these points is



unique. The ubiquity of this form of popular culture in this particular moment enables it to function as a new mythology, much as Barthes famously codified in his work on this subject in the 20th century. These arguments can go one step further: comics are now part of both political messaging and widespread myth-making.

Barthes's assertion that "myth is a language" (Barthes 2013, p. xi), and the ways in which the nuances of his argument connect to superheroes, are key for this analysis. When Barthes discusses myth as a language, he is thinking in terms of cultural mythologies, or rather in the ways in which clichés and culture collide. For him, myths are the stories humans tell ourselves and one another, more factually than narratively and more focused on identity than plot. In one particular example, Barthes discusses wrestling as a spectacle rather than as a sport (Barthes 2013, pp. 3–5). The movements are choreographed, designed for the spectators' enjoyment rather than the exertion of physical superiority over an opponent. Indeed, as Barthes points out, the audience of such matches does not care if the fight is fixed or choreographed. That is not their expectation, nor does Barthes think it should be as he delves further into the roles assigned by the performative world of professional wrestling. Andrew R. Bahlmann also considers Barthes, particularly his discussion of wrestling. Specifically, Bahlmann connects this commentary on the prescribed roles in wrestling to comic book superheroes, such as The Hulk (Bahlmann 2016, p. 11). The world of wrestling Barthes discusses, both in terms of its performative aspects and the clear-cut roles assigned to various characters, maps quite nicely, as Bahlmann comments, onto the world of superheroes. As will be demonstrated by my close reading of the characters in *Eternals*, there are particularly valuable insights to be gained by this connection.

Another scholar who considers Barthes's work significant in ways for this assessment of *Eternals* is Rebecca Houze. Houze references Barthes in that she offers "a study of how we read the images that surround us ... the ideas they signify" and the ways in which they are "located in graphic signage, in corporate identity systems, and, sometimes, in objects" (Houze 2016, p. 3). She takes the way in which Barthes considers myth to be a type of speech and offers a new way to see symbols and signs as new mythologies. In particular, she looks at the analysis Barthes famously offers on *Paris Match*, which he "demonstrated that the image on one level signified the soldier's patriotism, but on another reassured its white bourgeois audience of the validity of French colonialism in Africa, at a time of increasing political unrest" (Houze 2016, p. 2). This point of reference is useful in this analysis of *Eternals* as a new mythology, in that it simultaneously offers a rendition of Western canon and a postcolonial response to it. While certain aspects of the film seemingly celebrate and uphold the tradition of Western monopoly on literary and historical exceptionalism, other aspects of the film clearly aim to refute that narrative.

Another way in which Barthes's *Paris Review* can be considered is as a contemplation on the connection between mythology and empathy. The two levels on which the image can be regarded, movingly patriotic or approvingly colonialistic, speak to the ways in which perspective alters emotions. Arguably, it is the role of mythology to offer these different perspectives.

Lastly, Joseph Campbell considers the empathetic properties of myth from both a pedagogical and a sociological perspective. The purpose of myth, whether old or new, is to, as Campbell argues, provide a frame of reference through storytelling. These forms of repeated traditional literature take up residence collectively in the zeitgeist of a society. Consequently, "when the story is in your mind, then you see its relevance to something happening in your own life. It gives you perspective on what's happening to you" (Campbell 1991, p. 2). He asserts that "one of our problems today is that we are not well acquainted with the literature of the spirit" (Campbell 1991, p. 1). Although Campbell was writing 30 years ago about what he saw as a crisis in education: that Greek, Latin, and biblical literature used to be part of everyone's vocabulary, but that these fields are no longer ubiquitous, his argument is even more true today. The vernacular of popular culture in the 21st century has shifted; arguably, the cultural ubiquity of comic book superhero films makes them one of the strongest candidates to serve as new cultural touchstones.

When Campbell discusses myth's relationship to culture, he argues that these stories, "these bits of information from ancient times, which have to do with the themes that have supported human life, built civilizations, and informed religions over the millennia, have to do with deep inner problems, inner mysteries, inner thresholds of passage, and if you don't know what the guide-signs are along the way, you have to work it out yourself" (Campbell 1991, p. 2). Campbell's view of myth here lines up extremely well with the way in which *Eternals* functions as a modern mythology. Although the film is hardly ancient, the perspectives it offers of humanity, history, and morality are rooted in far older lessons and stories. The adaptation of the comic books, which is an adaptation of so much older literature, mythology, and history, shows just how much persists in society's efforts to describe, define, and seek a utopian ideal.

## 3. Results and Discussion

While considering the 2021 film, it is important to note that many of the value judgments and decisions present in the film were initially made over the decades since the 1970s through the medium of comic books. As with any adaptation, however, the degree of closeness to the source material and the choices as to what to change must be attributed to those creating the adapted work rather than the initial source material.

Marvel draws on classical, Western history and mythology in *Eternals*—in fact, all ten of the eternal deities have their roots in these sources. By making this creative choice, the film implicitly argues that this source material is foundational to humanity—more, that it is the very basis of humanity. Marvel adds a narrative twist to this argument: through the adaptation of the source material into a 21st century, diversely cast movie, *Eternals* takes the Western colonial narrative of the past and reshapes it into a postcolonial, optimistically utopian future.

This notion of using popular culture to speak to mythological traditions and to create new versions of them for the present and future is hardly unprecedented. *Star Wars*, *The Matrix*, and *Titanic* demonstrate how commercial success and popular culture hold stories such as these in the cultural zeitgeist, shared with generation after generation as beloved stories passed on much as oral tradition was thousands of years ago (Frauenfelder 2005, p. 211). Similarly, scholars such as Michael Bitz and Nancy B. Sardone have particularly considered the role comic books have in creating new pathways to both literacy (Bitz 2004, p. 575) and classical literature (Sardone 2012, p. 67). This discussion of *Eternals* draws on this scholarship in that this film models the ways in which this highly accessible medium can speak to 21st century issues and ultimately function as a new mythology.

What, then, is the mythology of *Eternals*? It is a seemingly contradictory combination of Western-centric lore and postcolonial readings, which, as is expected of mythological traditions, both answers basic questions of who we are and why we are here, but simultaneously asks how can we do better. By going through a close reading of each of the ten Eternals in terms of their source material, their adapted character, and the casting of the character in the film, it will be possible to consider the mythology they create with greater specificity. The ten characters divide easily into three sections: those who are based in Ancient Greek mythology, those of Mesopotamian origins, and those connected to British folklore, but not, perhaps, performed in the straightforward ways these categories seem to suggest, thus both subverting and adding new dimensions to this Western historical and literary tradition. As is clear from this categorization, the chronological, linear progression is undeniably, exclusively Western in terms of the source material. However, in many of the cases, the adapted character, both through the screenplay and especially the casting, have added nuance and complexity to this Western-centric, colonial narrative of the world.

Ikaris, Thena, Sersi, Ajak, Makkari, and Phastos all have roots in Ancient Greece. This choice to make the majority of the characters come from this single Western canonical root gives the impression from the very beginning that when these alien gods first arrived several thousand years ago, Ancient Greece was their primary focus, perhaps underscoring the Western notion that this was the cradle of civilization.

Ikaris is based on the Ancient Greek myth of Daedalus and Icarus. Daedalus, the master craftsman who built the Labyrinth, makes wings out of feathers and wax for himself and his son. However, he warns Icarus not to fly too close to the sun, which Icarus does anyway. The wax melts and he falls to his death. As with his namesake, Ikaris is able to fly. He can also fire energy blasts from his eyes, perhaps demonstrating some of the power that made his counterpart fall. Ikaris commits suicide by flying into the sun at the end of the film, another homage to the original myth. Richard Madden plays Ikaris in the film: a cis, straight, white male. He uses his natural Scottish accent rather than the colonialistic Received Pronunciation traditionally associated with British actors, perhaps mitigating the impression created by his face and body. By and large, however, Madden, and his Ikaris, conform to the colonialist image of the source material. Perhaps it is worth noting, however, that he is the villain of the story.

Thena is only missing the first letter of her famous Grecian counterpart: Athena. Her power again mirrors that of her antecedent. Thena is a fierce warrior, able to both wield and even create a variety of magical weapons. She also apparently suffers from Mahd Wy'ry, a mental affliction borne of living too long and seeing too much. It is discovered, however, that this illness is actually a sign of insight; she is able to perceive what is being concealed from the other Eternals. Consequently, despite the perception of it, Mahd Wy'ry is actually a sign of astute wisdom, fitting for a character named after Athena. Angelina Jolie plays Thena. Although she is American, her accent could perhaps be described as Mid-Atlantic. Regardless of the blend of British and America, she speaks English with an undeniably Western sound, lacking any of the mitigation provided by the accent Madden brings to Ikaris. She too, therefore, contributes to the colonialistic depiction of these characters in an even less diluted way.

Sersi, played by Gemma Chan, takes her name from Circe, a sorceress and minor goddess in Greek mythology, perhaps most famous for turning Odysseus's men into pigs in Homer's *Odyssey*. Sersi the Eternal wields a similarly transformative power, along with her species' innate immortality and durability. In one particularly dramatic moment, she stops a bus hurtling towards her by turning it into a gust of rose petals. Her power reaches its dramatic zenith when she channels the combined power of her fellow Eternals to turn the Celestial emerging from the center of the Earth to stone, consequently saving the planet. Sersi, in many respects, more accurately reflects the Circe of the *Odyssey*, who is intelligent, independent, and imminently more than a match for any man she meets. However, the character has often been read as a temptress whose magic serves as a misogynistic explanation for cheating, errant husbands, rather than as a sign of her creativity or power. Sersi, however, offers no room to be misconstrued. She is the undoubted hero of this film–she literally saves the world. The actress was born in England and her natural accent, which she uses in the film, reflects that. However, the actress's father is from Hong Kong and her mother is from China via Hong Kong. The line that leads Chan's genetic and cultural inheritance to perform the face and voice of a version of a Western classical mythical character is complicated, to say the least. The colonial history and postcolonial present which characterize the triangular relationship between the UK, China, and Hong Kong are embodied by Chan's Sersi. Her performance can therefore certainly be seen as a postcolonial response to the application of the source material.

Ajak is based on Ajax, the mythological hero from Homer's *Iliad*. During the Trojan War, Ajax was a formidable warrior, acting almost exclusively in defense of his fellow heroes. Ajak's power as an Eternal therefore makes sense: she is able to heal the others while also being a strong warrior herself as well as the group's leader. Selma Hayek plays Ajak, a Latina woman using her naturally Mexican-accented English. Moreover, this casting choice gender-flips the group's leader. These two choices considerably shift the colonialist narrative established by the Ancient Greek origins of these global, alien heroes. Taking a broad view of things, Spanish too represents a colonial presence in Latin America, which perhaps deserves a bit more attention given the emphasis the storyline places on the 1521

genocide of the Aztec empire by Cortés, but the attempt to offer a postcolonial pushback through this particular depiction of "Ajax" is clear.

Makkari is the Roman god Mercury, originally the Greek Hermes. As the messenger god, Mercury has winged that enable him to fly from one place to another at high speed. Makkari shares this gift: she is able to move at least as fast as her mythical counterpart. She is played by Lauren Ridloff, who is a Black American woman, and therefore, as with Ajak, gender-swaps the role. Ridloff's casting makes Makkari the first deaf superhero in the MCU; she fulfills the role of messenger primarily through sign language. As in the other cases discussed, there is a duality to the selection of Mercury as the inspiration for one of the ten Eternals and the casting of this role as a deaf Black woman. On the one hand, it is problematic to imply that the only way to give a deaf Black woman power is to have her character take its inspiration from the male Greco-Roman messenger god. On the other hand, this combination of creative choice and casting gives her access to a power and a level of name recognition she would not otherwise have with Western audiences. Deaf people are not usually cast as communicators; Black women do not usually get to play Mercury.

Similarly, Phastos is based on the Ancient Greek god Hephaestus, and as with his namesake, is able to craft tools and weapons and create technological advancements with his inventions. Played by Brian Tyree Henry, his performance adds another Black American to the narrative. Not only is he Black, but he is also the only one of the ten who is married with a child, and he is gay. His presence and his relationship with his husband caused the film to be banned in multiple countries after Disney refused to cut overt references to their relationship, such as kissing. Here is another powerful Ancient Greek deity, embodied by a Black American, and restoring something of a non-white demographic and perspective to the story. Again, however, the question as to why power has to bear a white, Western name and legacy is asked by this performance. With so many storied African traditions and deities, as discussed above with Makkari, why is it necessary to give this powerful, creative Black man a white name and a Western origin?

When taken all together, therefore, of the ten Eternals, six are drawn directly from Ancient Greek mythological sources. Consequently, this list inherently excludes many other ancient civilizations, such as the non-white and non-western ones found at the same time in places such as China, Egypt, or countless other possibilities. The narrative that places the focus of the Eternals on Ancient Greece further implies that this culture was more central to the development of humanity, more worthy of favor and attention, which unfortunately feeds directly into a trend of appropriating "the literature and history of ancient Greece and Rome to promote patriarchal and white supremacist ideology" (Zuckerberg 2018, p. 5). Consequently, the postcolonial responses inherent to the casting and performances of this film are worthy of even more scrutiny.

Moreover, the pattern created by these ten characters fits in with the ongoing conversation among scholars regarding the depiction of race in comic books. Marc Singer, Matthew Facciani, Peter Warren, and Jennifer Vendemia have considered the complex and tenuous relationship comic books have long had with depicting race (Facciani et al. 2015, p. 217). Singer identifies a shift that has taken place in this relationship in contemporary comics–one which he states is now "anything but simplistic" (Singer 2002, p. 107). Comic books, especially ones focusing on superheroes, traditionally trivialize and exclude characters based on race and gender, highlighting physical or sexual qualities and relegating non-white male characters to subordinate roles. These six characters in *Eternals* begin to shift this pattern, but there is clearly still a long way to go.

The postcolonial commentary inherent to this depiction of these characters, especially in terms of their casting and performance, contributes to the argument *Eternals* arguably offers: that through this particular retelling, a utopia is offered. Perhaps the film is not saying that humanity has arrived in that utopia, but that there is a potential for a utopia within the narrative and performance it offers. It is in the postcolonial casting of these roles that "a touch of irony" can be seen around the definition of utopia. While there is

a "realization that 'utopia' means 'no place' and so cannot be 'brought about'" (Bowman 2007, p. 74), there is clearly a desire to offer the audience one, even if it must inherently be imperfect.

The next category is Mesopotamian Eternals, specifically Gil and Kingo, who are references to two civilizations in particular: Sumerian and Babylonian. The assumption that myths "depict a quest or founding of civilization" (Brown 2017, p. 78) is important to keep in mind here as these two characters mark the first step in the developmental trajectory stemming from Ancient Greece. Moreover, while these civilizations are geographically rooted in the Middle East, and therefore are not necessarily of the same level of white, Eurocentric mentality, these two civilizations are still ultimately part of the Western narrative of the origin of civilizations. Consequently, the postcolonial adaptation of these two characters in the film is particularly important to understand.

Gilgamesh, also called Gil in the film, takes his name from *The Epic of Gilgamesh*. This epic poem comes from Sumer in Mesopotamia. Credited as the world's oldest, notable literary work and one of its oldest religious texts, this name places another of the Eternals at the historical heart of Western civilization and mythology. While Gil does not have any distinctive or unique skills, he is perhaps the most physically powerful of the group. Gilgamesh is played by Don Lee, whose original Korean, non-Westernized name is Ma-dong Seok. As a South Korean, Lee brings another piece of the Western colonial legacy. The complex history of South Korea, first with Japanese colonialism and then the partition under the influence of the US and Russia combined with South Korea's recent, extensive efforts to create a national media culture speak to the unique identity this character brings to the postcolonial composition of the Eternals.

Kingo's antecedents are less clear, but his namesake seems to be a minor Babylonian god. This identity connects him with Gil as a fellow Mesopotamian. Kingo's powers are not particularly distinct in the film, but he shoots energy out of his hands as a projectile weapon on multiple occasions. Kingo is played by Pakistani-American Kumail Nanjiani, a casting decision which brings further postcolonial connections to the ensemble. British colonialism in South Asia has a long, sordid legacy, compounded by the partition of India and Pakistan. Kingo's interest in Bollywood acting and his legacy as a performer in that industry over generations draws the viewer's attention to the character and the actor's shared cultural identity in this postcolonial context.

By casting these two characters, whose identities are integral in the Western narrative of the founding of civilization, as non-White, non-Western takes one step towards diversifying the new mythology built by *Eternals*. Specifically having the actors bring South Korean and Pakistani identities to the depiction of the characters makes this new mythology decidedly postcolonial. Consequently, the utopia offered by *Eternals* is a moving target. The postcolonial discourse implied by these two pieces of casting point to the "ever-shifting patterns" of "the contingencies of history and biography, which spin the kaleidoscope" around the "stimuli" (Litvin 2011, p. 58) of literature and mythology which ultimately build the postcolonial utopia offered by *Eternals*.

The final two Eternals, Druig and Sprite, are the youngest appearing characters, due to their casting, if not their characters' actual ages. This fact, perhaps not coincidentally, lines up with their less well defined and perhaps more modern identities. Neither Druig nor Sprite have clear, specific antecedents, but both have connections to British folklore, continuing the narrative of Western tradition embodied by the Eternals. However, "by incorporating folktales and modern folk beliefs" into "reconstructive efforts," it is possible to make "a systematic reassembly" (Roots 2020, p. 175) of a particular history and mythology. By choosing to incorporate these Western identities, even though they are not as specific as the others, Druig and Sprite pull *Eternals* towards an exclusionary construction of human history.

Druig is able to control the minds of others. He is played by the Irish Barry Keoghan, cementing the connection to the druids of Irish mythology and when spelled with a "g" at the end, is a cognate for "dragon" in British folklore. The fact that Keoghan retains

his naturally Irish accent in performance somewhat complicates the narrative of Western supremacy. Ireland has a difficult history with nationalism and imperialism, dating back to 12th century Norman military and political involvement. The modern division of Ireland and Northern Ireland has as recently as Brexit caused significant challenges. Thus, the history of colonialism, imperialism, and nationalism inherent to Irish history is embodied by Keoghan's Druig and his accented English.

Sprite is the third male character to be gender-swapped, along with Makkari and Ajak. She is played by the American Lia McHugh. In European mythology, sprites, from the Latin word for "spirit," sometimes have the power to manipulate various forms of matter or to trick unwary humans. Sprite the Eternal is known for the second of these gifts in particular. Given that her character does not place her as particularly as any of the other characters, the casting choice has even more weight. McHugh's Sprite is both American and young. The implicit argument here is that she is the responsibility of the other nine Eternals due to her youth, America can also be seen as the inheritor of these nine Western histories, literatures, and mythologies. Moreover, just as Sprite has taken part in the complete history of the Eternals on Earth, so too is America not absolved of its complex legacy of colonization and nationalism. Within McHugh's Sprite, therefore, is a depiction of the American identity, including pieces of both nationalism and a pushback against the Western legacy of imperialism.

Implicit in conversations regarding Western imperialism and explicitly outlined by some scholars, is the degree to which superheroes can be seen as deliberate militaristic and nationalistic propaganda. Scholars such as Mia Sostaric and Paul Hirsch have worked on these points, specifically thinking about the role of characters such as Captain America (Sostaric 2019, p. 20). This relationship was formalized in the Writers' War Board of World War II (Hirsch 2014, p. 448), but continues informally today. The presence and depiction of fictional governmental and military organizations such as S.H.I.E.L.D. and S.W.O.R.D., as well as the real world ones such as the USAF and the FBI, has frequently been pointed to as endorsement by both scholars and critics. This politically complex aspect of the field is particularly important when thinking about the nuances and problems associated with depictions of national, racial, and ethnic identity in *Eternals*. *Eternals* then in turn provides a perspective through which the audience can consider these depictions, both as problematic in some ways and utopian in others.

In key analysis of utopian texts, a pattern emerges in terms of the role literature, history, and mythology play in the process of world-building. In particular, utopian worlds use "fiction as their mode of representation" and that they simultaneously "displace their ideals into the past" and "project them into the future" (Bruce 2008, pp. xii–xiii). *Eternals* does all three of these things. It uses elements of traditional stories to build a world-wide, historical narrative. It casts the characters of this narrative both diversely and while paying special attention to postcolonial implications. Finally, it brings this narrative into the future it offers for the 21st century.

This compilation of casting and performance choices across these ten characters serves as a postcolonial commentary, enabling us to consider the political, social, and cultural implications of reading *Eternals* as a modern mythology. While myth has always played a vital storytelling role in human society, the nature and content of the myths humans write, tell, and share with one another necessarily shift over time. Standards change and expectations evolve, even as basic human needs remain the same. *Eternals* offers a new mythology, which speaks to this current moment and challenges the sociocultural expectations of its 21st century audience, which are paradoxically as monolithic and as diasporic as the American and international audiences measured in ticket sales. In particular, *Eternals* addresses current concerns about nationalism in our postcolonial landscape.

## 4. Conclusions

Given the financial success of the comic book superhero movie industry, it is safe to say that, for the time being, these films will continue to be a significant part of 21st century

popular culture. While one option is certainly to continue to relegate them to the realm of popular culture and low-brow entertainment, it is worthwhile to understand what is at the root of this popularity. It is worth understanding what brings so many people, both casual consumers and diehard aficionados to spend so much time, money, and care on these stories and characters.

*Eternals* walks a fine line, providing both familiarity and safe provocation. It honors and glorifies a Western narrative of literature, history and mythology that the audience has absorbed to various degrees. At the same time, it educates the audience in the ways in which this accumulation of narrative has shut out so many groups of people for so long. By broadening its narrative, the movie strengthens its heroes, finding a response in an audience that may lead them to ask why real life cannot reflect imagination. That it does this within the confines of three hours means that musings about what civilization is composed of may go on longer, but do not have to do so. And yet, to stimulate those ideas puts the movie squarely in the broader, higher conversation of what knits us together and what our myths should be now.

**Funding:** This research received no external funding.

**Institutional Review Board Statement:** Not applicable.

**Informed Consent Statement:** Not applicable.

**Data Availability Statement:** Not applicable.

**Acknowledgments:** My gratitude to Rich Layne and to Elizabeth Hull. I would also like to thank W.B. Worthen for his feedback on an early draft.

**Conflicts of Interest:** The author declares no conflict of interest.

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
