# Peer review of "Marvel Presents a Global Utopia and Confronts Nationalism: Eternals as a New Mythology Forged from Western Roots"

_humanities, doi:10.3390/h11030060_

Round 1

Reviewer 1 Report

In its present state, this manuscript needs a lot of careful consideration and substantial revisions before possibly getting to a publication stage. In terms of content, the analysis was very thin in nature and needed to add nuance and complexity throughout. For instance, thinking further about the production might aid the author(s) in their revisions. Marvel's casting director tends to be Sarah Finn so maybe a discussion of this could aid in unpacking the production decisions further. Additionally, the discussion of onscreen content was too glossed over--as such, it needs to be really fleshed out or omitted altogether as it currently does not add anything. As another suggestion, the literature review should have included further unpacking of Marvel films in general to allow the author(s) to better situate themselves into the existing conversations, especially in studies on media and popular culture.

In terms of flow and structure, the breakdown of the sections simply did not make sense. For instance, the first section was the "Introduction" and then immediately following that was the "Results and Discussion" section. This was confusing as we know that the literature review should go right after an introduction to establish the existing conversations to allow authors to  better situate their original contributions and own voice into the existing context. It turned out that over 60 percent of this "Results and Discussion" section was focused on existing literature. The analysis then seemed like an afterthought in this same section. Exacerbating this, the "Conclusion" section did not offer any aid as it was less than 50 words in length. As such, this last section needs to be carefully considered and the "so what" factor needs to be better addressed throughout the manuscript for readers.   

In terms of formatting, this manuscript needs heavy editing. For example,  the in-text citations for direct quotations did not include page numbers but needed to. Additionally, the in-text citations were placed outside the periods (and while the exception for this tends to be for block quotes, none were used in this manuscript). There were also, many, many typos--including misspellings of several referenced scholars such Barthes, Campbell, etc. 

Author Response

Hello,

Thank you very much for your feedback.  Please see my attached revision.

Best regards - 

Reviewer 2 Report

This article offers up a decent working through of the recent Marvel film Eternals, but one that perhaps could do with the some finetuning to really bring out the merits of the argument. The article interrogates the question of mythology and the MCU, drawing on important scholarship and textual analysis of its target films to establish how Eternals provides audiences with “a classical western canon remade as a multicultural utopia.” There is plenty to commend here and I enjoyed the points raised, but upon reading the piece I felt it needed another draft to pull out the “political, cultural, and social” contexts that frame the film under discussion. One idea to refine the prose would be to strip back and rework the introduction, cutting out the signposting and narration regarding the film as “objectively good or bad” or whether it should have made more money. Instead, there is space to explain the film in more detail – its narrative, but also its co-option and treatment of a new mythology drawn from Classical Mythology and “Western canonical texts”. Set up the argument, and ‘show’ but do not ‘tell’ – I think the introduction does not quite advertise the strengths of the piece, and could go further. Some of the section transitions in these early stages (as throughout) could also do with smoothing over (see the beginning of section 2 “Results and Discussion”), and I spotted instances where paragraphs run to only a sentence, which creates a staccato effect to the prose.

From a conceptual standpoint too, the risk with writing on Marvel is effusive praise towards the films themselves – I would encourage the author to cut out instances that read as behind-the-scenes trivia (such as the passage discussing Iron Man’s final line or the terminally ill child from Thor: Ragnarok). There also needs to be a more rigorous understanding of ideology, particularly as the author uses “good” and “bad” in generalised ways, particularly loaded given the historical and cultural connotations of such terms. Such observations equally rehearse industry narratives around the films, when cutting this back leaves more space to discuss the prominence and relevance to Eternals, which can get a little lost. On that note, the other issue is a structure that again does not quite to justice to the arguments around the film being presented. For example, after the introduction, we do not return to Eternals until page 5 (of 8), and instead the opening half is a broader working through of the mythmaking of Marvel by examining a cross-section of exemplary films, characters, and narratives. This analysis offers a solid survey of morality and value judgments across the MCU, and so I wonder if this needs reversing, and after the introductory paragraph we should go straight in with Marvel to keep them as the centre of the piece, setting up Eternals’ place within this history and then getting into the wider critical paradigms. This means that after the MCU (its phases, narratives, spiralling stories), the author can then bring in Roland Barthes (check spelling of this name through – especially page 2, line 60) and myth, classical mythology, and popular culture to ‘make sense’ of Marvel as a multimedia phenomenon, and enforce the ways that the MCU represents a new “canon” of stories that can be understood as myths. As it stands, we instead begin with long accounts of Barthes, Campbell and Fauenfelder, and it takes a while for the connections to the MCU, and later Eternals, to become clear.

A few others points to consider:

- Italicise all film titles, and ensure all quotations have page numbers (if required by the journal).

- reference the fact these stories are taken from comic books and are, as such, adaptations.

- the MCU has a rumoured 'business' relationship with the U.S. government/military, and so its “value judgments” are ideological and highly problematic because representations are naturally coded around nationalist narratives and constructions of heroism etc. (see media pushback on this: https://thedirect.com/article/marvel-movies-military-us-influence-mcu-james-gunn). It would be good if the author therefore complicates the observations around the MCU here – essentially, to what extent should we be celebrating these films?

- the analysis of Eternals is astute in how it identifies cultural and “postcolonial overtones” in its narrative but, as described above, comes too late on. It would be nice too to have more on the director Chloe Zhao, critical responses to the film, casting (of a number of Western/Hollywood actors), progressive representation and the issue of multicultural utopianism. Again, at times the analysis reads like a list of characters, features and interesting observations, when such points could more robustly be tied to a central thesis.

- cut out any instances of colloquial language (“vibe”, “and so on”, “woman of color, no less”).

- cut out long book titles when introducing the scholarship.

Author Response

(The authors gave the same response as above.)

Round 2

Reviewer 2 Report

This article is, I’m happy to say, much improved, and I commend the author for their careful reworking and refining of the piece on the Eternals in ways that really bring out the specificity of the argument and its relationship to myth and morality. The introduction does a much better job of outlining the scope of the article, and identifies the main critical paradigms with greater economy. I would say that, given the additions, it does run a little longer (before we get to the second Literature Review section). If there are opportunities to cut back some of the prose prior to this point, then losing a paragraph might be helpful. Anything cut here can be added to the second section in some abridged form anyway, which again is altogether more efficient in outlining the intersection between comic books, superheroes, myth, and identity politics. The author might want to cut the reference (bottom of page 4) to the first mention of Bahlmann, as that keeps the argument focused on Barthes without a small deviation to the work of another scholar. Bahlmann can then be simply introduced in the subsequent paragraphs on page 5.

A few other thing I spotted:

- “ensemble cast” rather than just “ensemble” (p.1)

- cut out the passive voice (“it can be seen”)

- film information needed (director, year of release)

Overall, I would perhaps still like to see more stripping back of the more diffuse sections and paragraphs – this is easily done doing the final revision stage, as there are numerous sentences that appear to be the author plain speaking, so any explicit signposting can be smoothed out. This is because it is not until page 7 that we get into the analysis proper of Eternals. If the author can cut a page from everything before this, then there will be a better balance between contextual concerns/critical paradigms and close textual analysis. In fact, cutting out some of the earlier stuff would allow you to re-integrate it later on, as the citations notably reduce once the film takes centre stage. To avoid the piece reading as top-heavy, I would suggest synthesising some of the critical material in the first half gradually into the second (again this should be easily done).

Author Response

Thank you very much for your feedback.  I have made changes in a Word document using track changes as requested.  I have removed the frequent use of first person throughout the article.  I have shortened the introduction and streamlined it in order to get to the meat of my argument more quickly.  I have moved some of the critical references out of section two and incorporated them into relevant parts of section three.  I removed the passive voice--I think I got it all.  I have tried not to sign post and to generally condense.  Please let me know if I missed anything or if anything is still lacking/insufficient.  I would be happy to revise further as needed!